# Cloud-Based Collaborative Road-Damage Monitoring with Deep Learning and Smartphones

**Akshatha Ramesh [1], Dhananjay Nikam [2], Venkat Narayanan Balachandran [2], Longxiang Guo [2], Rongyao Wang [2], Leo Hu [2], Gurcan Comert [3] and Yunyi Jia [2,\*]**

[1] Department of Electrical and Computer Engineering, Clemson University, Clemson, SC 29634, USA; akshath@g.clemson.edu

[2] Department of Automotive Engineering, Clemson University, Greenville, SC 29607, USA; dhanann@g.clemson.edu (D.N.); vbalach@g.clemson.edu (V.N.B.); longxig@clemson.edu (L.G.); rongyaw@clemson.edu (R.W.); leominghu@gmail.com (L.H.)

[3] Computer Science, Physics and Engineering Department, Benedict College, Columbia, SC 29204, USA; Gurcan.Comert@Benedict.edu

\* Correspondence: yunyij@clemson.edu; Tel.: +864-283-7226

**Abstract:** Road damage such as potholes and cracks may reduce ride comfort and traffic safety. This influence can be prevented by regular, proper monitoring and maintenance of roads. Traditional methods and existing methods of surveying are very time-consuming, expensive, require a lot of human effort, and, thus, cannot be conducted frequently. A more efficient and cost-effective process is required to augment profilometer and traditional road-condition recognition systems. In this study, we propose deep-learning methods using smartphone data to devise a cost-effective and ad-hoc approach. Information from sensors on smartphones such as motion sensors and cameras are harnessed to detect road damage using deep-learning algorithms. In order to give heuristic and accurate information about the road damage, we used a cloud-based collaborative approach to fuse all the data and update a map frequently with these road-surface conditions. During the experiment, the deep-learning models achieved good prediction accuracy on our dataset, and the cloud-based fusion approach was able to group and merge the detections from different vehicles.

**Keywords:** road damage; machine learning; CNN; LSTM

## 1. Introduction

One of the causes of the U.S. road accidents today is road damage [1]. Road problems such as potholes and iced-over stretches of highways cause many traffic accident deaths each year. Well-maintained and monitored road surfaces could increase road-user safety, fuel economy, and comfort levels. The currently widely used methods for monitoring road-surface conditions include using profilometers [2] and surveying techniques. Surveying is the traditional road-condition monitoring and inspection method that requires a technician to walk or drive along the roads to search for defects manually. Such a process requires a lot of human and equipment effort, and can still hardly provide timely information about road damages. In addition, the profilometer equipment is expensive and requires trained professionals to operate it, making this process costly and impossible to execute frequently.

To simplify monitoring processes by lowering the requirement of special equipment, smart detection and classification algorithms have been developed. A significant number of those methods are based on vehicle motion signals such as speed and acceleration. Another important category of those methods is vision-based detection. These recently developed methods are much more convenient than the traditional methods for detecting road damage, but they still rely on dedicated data-collection devices, which limits the scale of their applications. Thus, if an effective and cost-efficient method is developed, it would be beneficial to monitor road damage continuously to enhance the transportation system in

terms of driving safety and comfort. The usage of smartphones in scientific research has increased exponentially in recent years. Smartphones come with increasing computing power and a great variety of sensors, such as accelerometers, gyroscopes, magnetometers, GPS, and cameras, and many people are carrying smartphones with them when they are driving. These features make using smartphones as monitoring nodes possible.

Thus, the objective and contribution of this paper are to develop a cost-effective approach to monitoring road damage, including potholes and cracks, using in-vehicle smartphones. The proposed method consists of both vehicle-motion-based and vision-based road-damage detection and classification methods to improve detection accuracy. The method also contains a cloud-based fusion algorithm to fuse all road-damage detection results from different vehicles in order to provide holistic, accurate and complete monitoring of road damage.

The rest of the paper will be organized as follows: Section 2 briefly reviews related existing research. Section 3 introduces the motion-based and vision-based road-damage detection methods and the cloud-based fusion and severity-estimation method. Section 4 introduces the data-collection process, the experimental results, and the analysis. Section 5 concludes the paper, and discusses the limitations and future work.

## 2. Literature Review

Many research efforts have been devoted to the field of the smart monitoring of road conditions or road damage. This section will briefly review existing work that employs motion-based and vision-based detection methods, as well as cloud computing technologies.

### 2.1. Related Work

The mainstream of smart road-damage/condition monitoring consists of motion-based methods that rely on vehicle speed, acceleration, and position data to make predictions on the status of the road surface. For example, [3] discussed a transfer function-based method to estimate power spectral density using the relationship between the road surface and vehicle acceleration and then estimate road-damage levels. The authors of [4] developed a piece of dedicated equipment called CarMote to monitor road-surface conditions. The device can measure accelerations and send data to roadside units. The authors of [5] designed a mobile sensor network using a public transportation system called BusNet. The system uses acceleration sensors mounted on buses to monitor the conditions of road surfaces. The authors of [6] used the statistics of vertical acceleration to detect potholes and bumps on roads. The authors of [7] applied thresholds to the value and the changing rate of vertical acceleration to detect potholes, and achieved a precision of up to 89%. The authors of [8] also used the changing rate of vertical acceleration to detect potholes, and the detection rate was around 80%. The authors of [9] trained a support vector machine (SVM) to detect traffic conditions based on a vehicle's longitudinal acceleration and obtained a false positive rate of 2.7%, and a false negative rate of 21.6%. The authors of [10] also employed SVM to detect potholes and bumps based on the standard deviation, mean, variance, and other features of a vehicle's vertical acceleration over 2 s; the false positive rate was 3%, and the false negative rate was 18%. In recent years, the usage of deep-learning models has become popular. The authors of [11] used both a convolutional neural network (CNN) and long-short term memory (LSTM) network to detect potholes, speed bumps, street gutters, and some other stability events using vehicle vertical acceleration, and achieved an accuracy of 93% for CNN, and an accuracy of 82% for LSTM. The authors of [12] used both CNN and LSTM to detect type of pavement based on the time and frequency domain features of vertical acceleration and achieved about 91% accuracy for LSTM, and 93% accuracy for CNN. This work also demonstrated that the placement of the inertial sensor (below suspension, above suspension, and near the dashboard) does not influence the detection accuracy by more than 2%. The authors of [13] analyzed which factors affect the acceleration measurements by smartphones in a moving vehicle the most and how they change road-roughness measurements.

Another major trend of existing road-surface condition monitoring is vision-based methods.The authors of [14] discussed using a CNN-based network for the detection of road-distress type using data from a specialized setup of cameras. The work by [15] evaluated the different segmentation algorithms for the pavement distress type. In [16], quantifying the distress type was considered along with identification and classification. Furthermore, the work by [17] discussed a deep fully convolutional model for crack detection (CrackPix), which leveraged well-known image-classification architectures for dense predictions by transforming their fully connected layers into convolutional filters. In recent years, the Road Damage Detection and Classification Challenge appeared and has drawn much attention. The authors of [18] used faster regions with convolutional neural networks (R-CNN) to detect eight different types of road damage in the challenge and achieved a mean F1-Score of 0.6225. The authors of [19] used a one-stage detector called "You Only Look Once" (YOLO-v4) for the challenge, and achieved an F1 score of 0.628. The authors of [20] used a variant of Fast R-CNN, the bi-directional feature pyramid network (BiFPN), and achieved an F1 score of 0.6455. Other participating teams also used variants of YOLO [21,22] or R-CNN [23,24], and their F1 scores were between 0.5 and 0.66.

Cloud computing technologies have also been used to help the monitoring of road damage or conditions. The authors of [25] offloaded heavy computation to a cloud database to achieve the faster and more precise detection of road conditions on smartphones. The authors of [26] used road-condition data saved on the cloud to generate alerts for end users. The authors of [27] ran a fast unsupervised road-damage detection method on edge devices, and a slower but more precise machine-learning model on the cloud to realize fast, real-time detection and precise warning of road damage at the same time.

### 2.2. Challenges and Gaps

Traditional monitoring and inspection of road conditions require surveyors to go along the roads to search for defects. Such processes are very time-consuming, expensive, and labor-intensive. Existing automated road-condition monitoring research has been studying motion-based and vision-based detection approaches. However, such approaches usually require special vehicles equipped with specific sensors associated with complex processing systems, which is still time-consuming and expensive. In addition, most of these approaches cannot provide real-time and in-time monitoring of road conditions. Although some recent research efforts have preliminarily investigated the possibility of using smartphone sensors for a particular type of road-condition detection [11,13], they usually use just one type of sensor, such as a motion sensor or a vision sensor. Few have used both motion and vision sensors for detection. Furthermore, these approaches are all at the single-vehicle level, without the leverage of multiple vehicles for a more holistic and in-time detection. Furthermore, the identification of road-damage severity has not been investigated as a further detection of damage types. Therefore, a more time-efficient and cost-effective approach to holistically monitoring the real-time and in-time road conditions with damage severity is needed.

This paper proposes a novel cloud-based collaborative road-damage monitoring approach using multi-sourcing in-vehicle smartphones. It leverages a larger number of existing common vehicles on roads and uses both onboard motion and vision sensors from smartphones to conduct the road-damage monitoring in a more cost-effective and time-efficient way. Furthermore, it leverages the multiple sources of detection from different vehicles with smartphones and fuses them in a cloud to achieve a more holistic detection of road conditions, including damage severity.

### 3. Cloud-Based Collaborative Road-Surface Monitoring

This section introduces the details of the proposed cloud-based collaborative road-surface monitoring method. The first subsection will present the framework of the proposed method. The second and third subsections will describe the vision-based road-condition detection using CNN, and the motion-based detection using LSTM. The fourth subsection

will give details on the cloud-based fusion algorithm for detection results from different networks and vehicles. The fifth subsection will explain how the severity of the damage can be estimated using data on the cloud.

### 3.1. Framework of Cloud-Based Collaborative Road-Damage Monitoring Method

Modern cellphones are packed with advanced sensors, including a magnetometer, accelerometer, gyro, and GPS, which enable them to measure precise acceleration and speed. When vehicles are driven on roads with different surface conditions, cellphones will receive acceleration stimuli with different patterns and amplitudes, and such different patterns can be used by machine-learning models to detect and classify road damage such as potholes and cracks. Modern cellphones are also equipped with high-quality cameras that can directly 'see' the surface condition of a road when the vehicle is driven on it. Moreover, many people carry their cellphones every day when they travel, which makes mobile phones very suitable for forming a sensor network that monitors road damage continuously: as long as enough people are using the road-damage detection app that is going to be introduced in this paper, the timeliness of road-damage information can be assured.

In this paper, the proposed cloud-based collaborative road-damage monitoring method utilizes both motion and vision data from cellphones to achieve more reliable and precise detection of road-surface conditions. The framework of the method is shown in Figure 1. The method contains three major modules: the local detection and prediction module, the cloud-based fusion module, and the user interface (UI). The local detection and prediction module includes dedicated deep-learning models for evaluating vehicle-motion input and vision input separately. The motion and vision-based detection results are shared to a cloud database through cellular communication. The cloud database will receive multiple detections from different vehicles passing the same area over time, and the cloud-based fusion module will fuse all those detections to generate a map that contains highly credible road-surface condition information. The UI will show users the road-damage map via a cellphone application and warn the driver of potential danger. Figure 1 is the overview of the proposed approach.

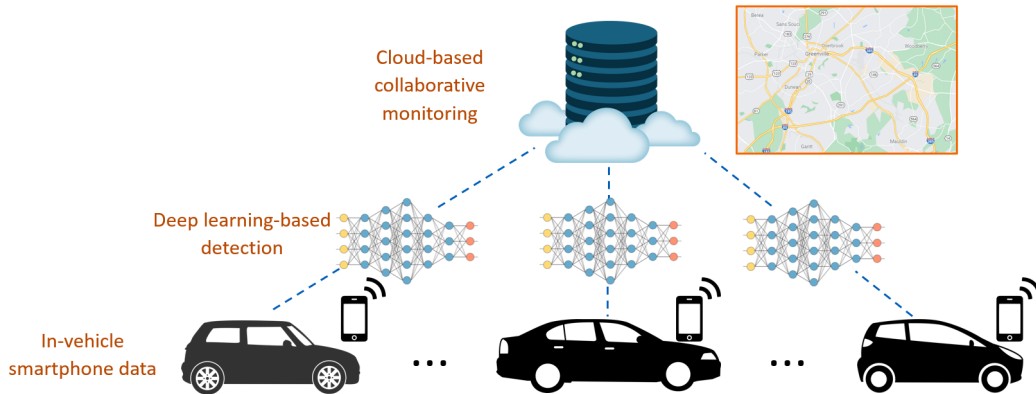

**Figure 1.** Cloud-based collaborative road-damage monitoring using in-vehicle smartphone data and deep learning.

### 3.2. Motion-Based Road-Damage Detection

Sensors on smartphones such as accelerometers, gyroscopes, g-force, and magnetometers can reflect the profiles of the road when the smartphone is mounted in a car that is driven on the road. One of the recurrent neural network (RNN) variants, LSTM [28], is used in this paper to utilize these motion data for road-surface classification. The reason for selecting the LSTM network is that, unlike standard feed-forward neural networks, LSTM has feedback connections. It can process single data points (such as images) and entire sequences of data (such as speech or video). This specialty is critical for our method, since

a single acceleration or speed data point cannot indicate a certain type of road-damage condition. The data are meaningful only when they are in context.

The proposed method utilizes an architecture that is a stack of two fully connected layers and two LSTM layers. Figure 2 shows the architecture of the proposed architecture. The first fully connected layer has an input size of 200 hidden units, followed by another hidden layer. The two LSTM layers follow this, with 64 units each stacked on each other. An LSTM unit in the LSTM layers is composed of a cell, an input gate, an output gate, and a forget gate. The cell remembers values over arbitrary time intervals, and the three gates regulate the flow of information into and out of the cell. The first gate is called the forget gate and it selects the part of the cell-state data to be removed. The second gate is the input gate, and it determines the data to be added to the cell state. Lastly, the output gate generates output data using the cell state. The output from the last cell is taken out, and a softmax function is applied to it. The final output is the probability of each class the model is predicting.

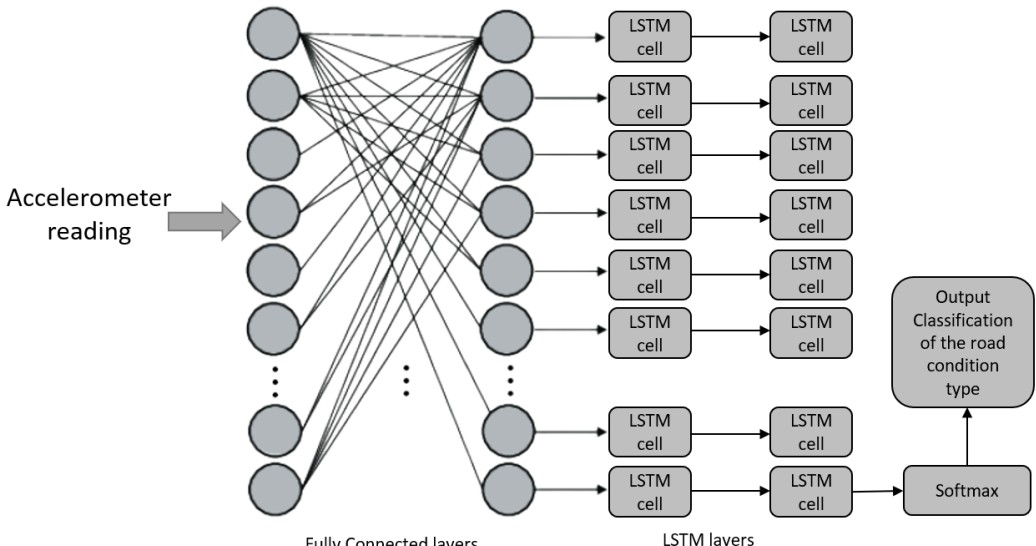

**Figure 2.** LSTM model structure for motion-based road-damage detection.

### 3.3. Vision-Based Road-Damage Detection

Detecting road damage is one subdivision of an object-detection problem, which is a process that utilizes computer vision and image processing to deal with detecting instances of semantic objects of a certain class in digital images and videos. Among all the object-detection methods, machine learning, especially deep convolutional neural networks, is the most prominent. Various network architectures, such as R-CNN, YOLO, R-FCN, and SSD, have been proposed. We selected YOLOv5 [29] and used transfer learning on it to achieve the detection of potholes and cracks. The training images were labeled by us to mark the areas of potholes and cracks, and the YOLOv5 model will catch the features of each road-damage type during training. The trained model will take the live video captured by the smartphone's camera as input and output bounding boxes and labels at the predicted areas of potholes and cracks in each frame.

In YOLOv5, the Leaky ReLU activation function is used in the middle/hidden layers, and the sigmoid activation function is used in the final detection layer. The default optimization function for training is SGD. YOLO works by first dividing the image into grids, and, for each grid cell, it simultaneously produces bounding boxes, confidence, and class probability. The model then aggregates those results to make the final bounding box output and classification. This architecture is known for both its performance and efficiency. Over the years, there have been multiple iterations and improvements over the original YOLO architectures. This latest iteration, YOLOv5, utilizes cross stage partial network (CSPNet) as the model backbone and path aggregation network (PANet) as the

neck for feature aggregation. These improvements have led to better feature extraction, and a significant boost in the mean averaged precision score.

### 3.4. Cloud-Based Collaborative Fusion

Cloud-based fusion is needed for the proposed method due to two reasons: Firstly, one vehicle cannot easily cover the entire road network, or even all road damage in one pavement section, during daily driving. However, the chances of covering all road damage and the entire road network increase if enough vehicles run the detection algorithm. A server on the cloud is required to gather this multi-vehicle data. Secondly, both vision-based and motion-based detection methods will unavoidably report false positives and false negatives, and such misleading information can be filtered out using cloud-based fusion. The type of damage and the GPS coordinates at which the damage is detected are used for the cloud-based collaborative fusion. The damage reported are stored in a relational database in the cloud. The relational database has GPS coordinates (i.e., longitude and latitude), type of damage, and confidence as its attributes. Considering there will be multiple entries for the same location and reporting damage, there is a need to consolidate this reported damage, and this need is fulfilled by the cloud-based fusion. The complete architecture of this approach is shown in Figure 3. Amazon web services' (AWS) Lambda function is used with AWS API gateway for posting the data to the relational database. The raw data in the database is optimized using clustering techniques to populate another database with optimized data. Using the AWS Lambda function and AWS API gateway, this data is posted to the website. The clustering technique used is explained in detail in Section 3.

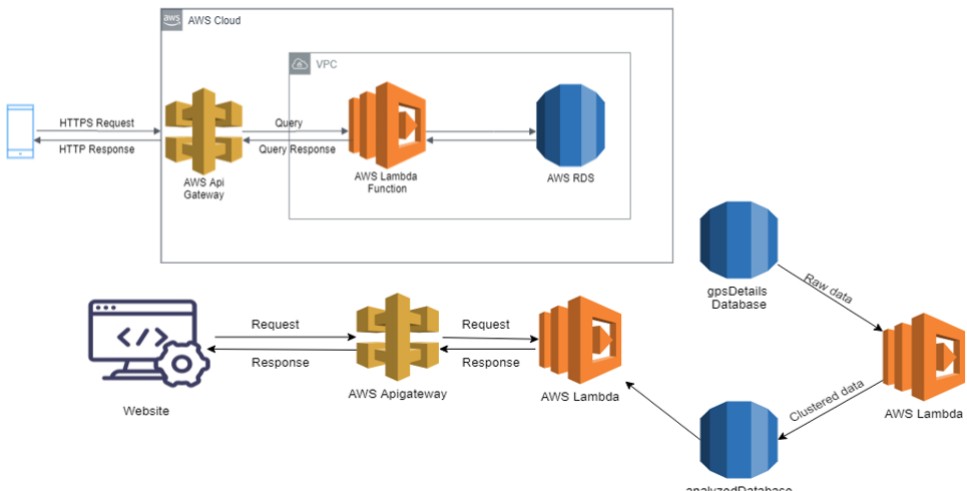

**Figure 3.** Architecture of cloud-based collaborative fusion.

Figure 4 gives a brief description of how the optimization of data is carried out on the reported damage. Different reports made at the same location might not have the exact same GPS coordinates, resulting in the database having a huge amount of duplicated damage reported for the same location.

With the need to consolidate these reports, an unsupervised machine-learning algorithm, k-means clustering, is implemented on the GPS coordinates. K-means is a numerical, unsupervised, non-deterministic, and iterative method [30]. Its efficiency and effectiveness in clustering have been proven in many practical applications. The algorithm starts with randomly selected *k* centers, where the value of *k* is pre-selected [31]. The algorithm then assigns each data point to its nearest center. The nearest center is generally calculated by using the Euclidean distance. This creates the initial *k* clusters [32]. After all the data points are assigned to their respective nearest center, the algorithm recalculates the centers. The new centers are calculated by averaging the data points that have been assigned to the initial centers. These recalculated centers are the new centers and the algorithm goes

through all the data points again to assign them to the new nearest center. This recalculation and reassignment are repeated until the criterion function becomes minimum or the algorithm has looped for a predefined number of times.

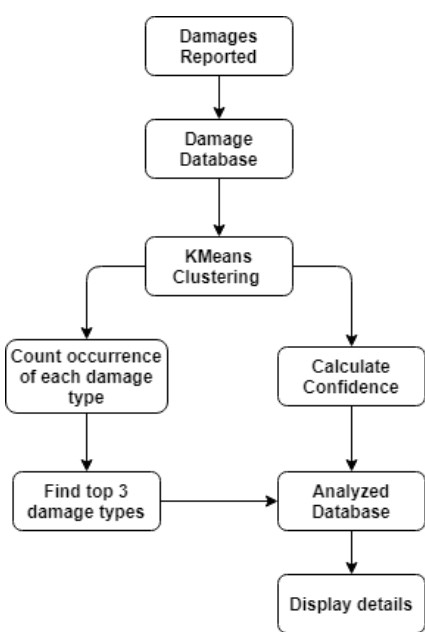

**Figure 4.** Cloud-based collaborative fusion flowchart.

In our approach, we use only the longitudes and latitudes of the reports from the data points for clustering. We start the implementation with an initial value of $k$ as 1 and perform k-means clustering on the entire data set. Within clusters sum of squares (WCSS) is the measure of the average squared distance of all points within the clusters to the centroid of the cluster and is calculated for all $k$ clusters; then, the average WCSS is calculated from the $k$ WCSSs. Next, we increase $k$ to $k + 1$ and repeat the process, until the difference between the average WCSS of the current $k + 1$ clusters and that of the previous $k$ clusters is less than or equal to 0.001. Once that condition is reached, the number $k$ is the optimum number of clusters that can be created given the current data set. The $k$ centroids generated by applying k-means clustering are the longitude and latitude values representing the data points assigned to that particular cluster. The data points present in each cluster have a very high probability of being at the very same location or within a very small distance from one another.

We consolidated the entire data set of n rows into k clusters where k is less than n. For the k clusters to truly represent the entire data set, we need to analyze the data points in each cluster so that, along with the centroids representing the location of the cluster, the damage type and confidence can also be represented by the cluster. In order to achieve this, we segregated the data points according to the clusters they were assigned to. Each cluster and the associated data points were then visited to find the type of damage reported, and the occurrence of each type of damage in the cluster was counted.

The top three most occurring types of road damage within a cluster are the most possible or significant ones for that particular cluster. Then, we could represent the cluster location with the centroid values of that cluster and the most significant damage types by the three most occurring damage types, and we need to find the confidence for each of these damage types. This was performed by calculating the sum of confidence of each corresponding damage type and dividing it by the total number of occurrences. The process of finding the types and confidence was performed on all the k clusters. We, further, stored the analyzed information of the clusters in a cloud database. The database consists of the longitude and latitude, the centroid of the cluster, and the corresponding top three types

of damage and their confidences. Additionally, we stored the cluster ID to distinguish different clusters in the database. This workflow is explained above, as shown in Figure 5.

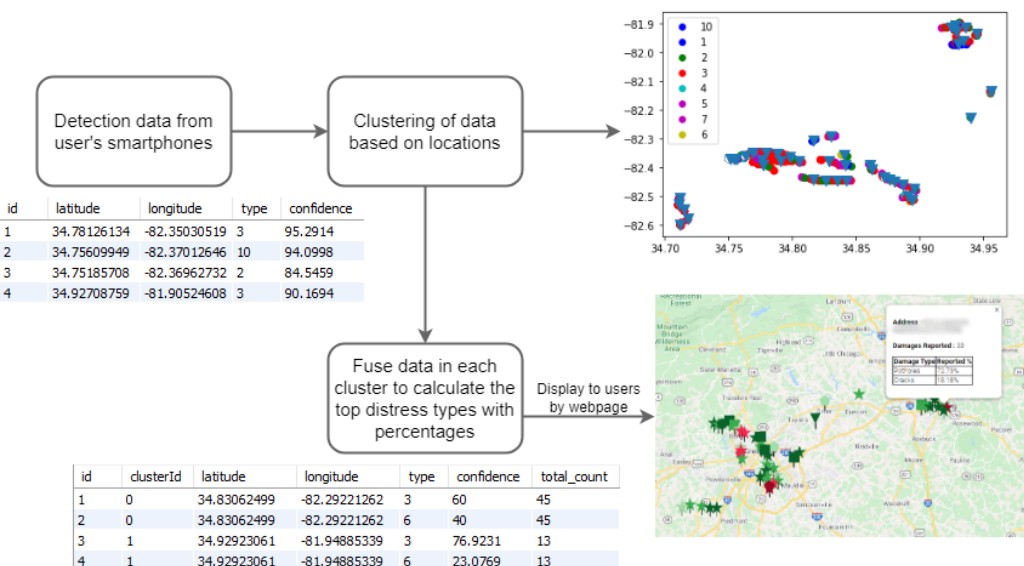

**Figure 5.** Back-end process of cloud-based collaborative fusion.

### 3.5. Road-Damage Severity Estimation

Apart from utilizing vehicle-motion data to detect the existence of road damage, we also utilize motion data to estimate the severity of the road damage to obtain a comprehensive monitoring of road damage.

#### 3.5.1. Method Design and Data Acquisition

Generally speaking, the most direct impact from road damage is a vehicle bump, and a vehicle's bump can be reflected by vehicle vertical acceleration. On the other hand, vehicle vertical-acceleration change is mostly caused by vertical road-profile change. Therefore, it is important to detect if there exists any relationship between vehicle vertical acceleration and road-damage depth.

To inspect whether there is a reasonable relationship between vertical acceleration and road-damage depth, we collected the historical data on vehicle vertical acceleration on different road-damage sections. For each road-damage location, data from multiple runs were collected. In addition, we measured the length, width, and depth of road damage using tape measures. Once all data samples were collected, the next step was to find out the useful metrics. In this case, we used the different combinations of maximum value, average value, and variance to investigate what leads to a reasonable prediction of road-damage severity levels. Take pothole as an example, assume we have $M$ potholes of different sizes, and, for each pothole, collect acceleration data of $M$ separate runs. Firstly, we find the average and maximum value of each run, which lead to an $M$ maximum value and $M$ average value. For both $M$ maximum value and $M$ average value, we then further calculate three statistics: their maximum value, average value, and variance, so that each pothole has six values to pair with damage depth. The chart is shown in Figure 6.

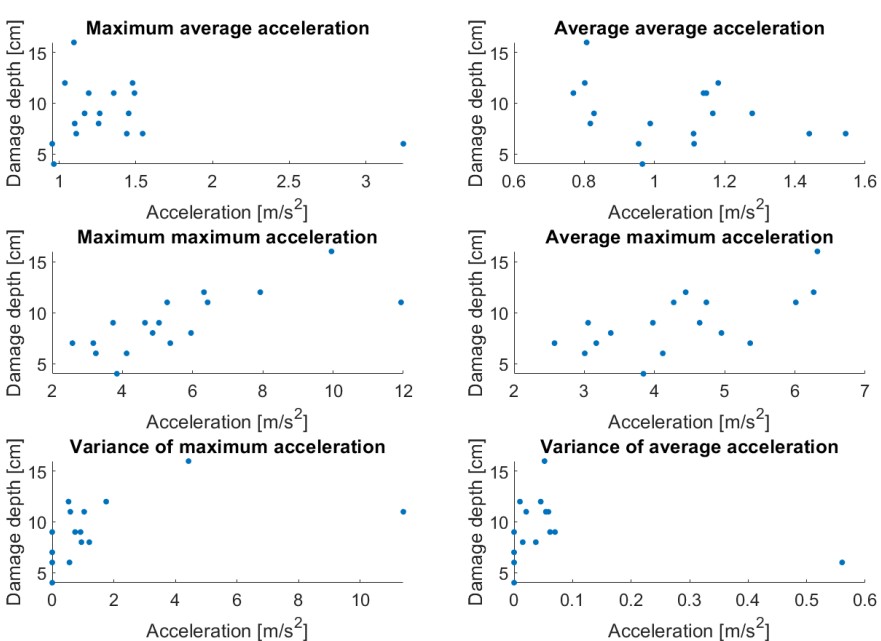

**Figure 6.** Data visualization of different road-damage estimation parameters.

Among all the combinations, the maximum of all the maximum accelerations data depicts a close-to-linear relationship to the depth of road damage. Thus, it is possible that the worst-case vehicle vertical acceleration can be used to estimate the road-damage severity. An example of a linear fit of data collected from 15 locations is shown in Figure 7.

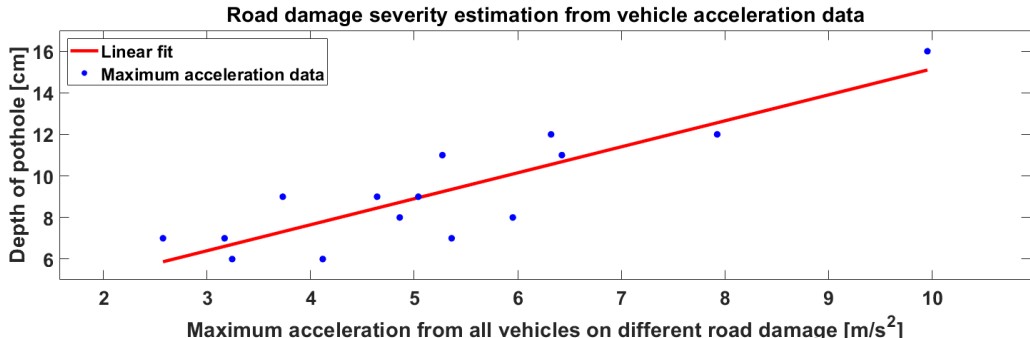

**Figure 7.** Road-damage severity estimation based on acceleration data in the worst case scenario.

### 3.5.2. Cloud-Based Road-Damage Severity Estimation

Taking the line fit result as a reference, the road-damage severity (depth, in this case) can be found from the worst-case vehicle vertical acceleration, which needs to be obtained from multiple different acceleration trajectories collected from the same location. Such different trajectories are generated by different vehicles running over the road damage at different speeds or different approaching angles.A single trajectory from one vehicle cannot be the basis for a valid prediction since the vehicle may run over the road damage at different speeds, which results in different acceleration trajectories, or even not run over the road damage at all. Consequently, the aforementioned estimation method, which is based on the data of multiple vertical-acceleration trajectories, can only be achieved using cloud-based architecture so that the algorithm can have access to data from multiple vehicles. The cloud-based road-damage severity estimation process is shown in Figure 8. The process leverages the cloud server described in the previous section, and the vertical-acceleration trajectories of a vehicle will be sent to the cloud together with the prediction result. Data collected from the same location will be grouped based on the location using the k-means

method. The maximum vertical acceleration of all available data is used to estimate the damage severity on that road section; thus, the estimation reliability increases as the size of the dataset increases.

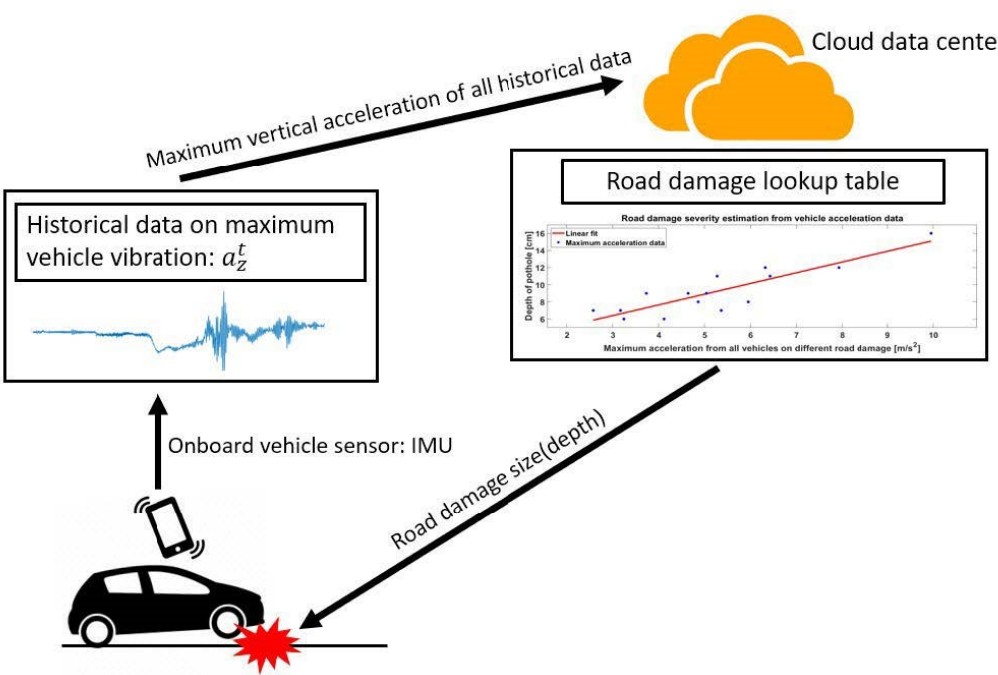

**Figure 8.** Cloud-based road-damage severity estimation process.

## 4. Experiment Results

In this section, the data-collection process and experimental results will be presented in detail.

### 4.1. Experimental Setup and Data Collection

To conduct the experiment, we developed a dedicated data-collection app, which will be introduced in Section 4.1.1. The composition of the data collected using the app, as well as the data that are available from the existing data set, are presented in Section 4.1.2.

#### 4.1.1. Setup and Application Development

This section includes a data-collection app [33] and explains how the data were collected using a smartphone from the sensors in smartphones. An existing android app was modified, which records the data from sensors—accelerometer, magnetometer, gyroscope, and GPS—along with the video of the roads where the data was being recorded, simultaneously. The UI of the app is as shown in Figure 9. The app gives two outputs—a video file and a .csv file that contains the sensor readings. During both data collection and real-time detection, the smartphone with this app is mounted onto a windshield, as shown in Figure 10.

#### 4.1.2. Data Collection

The smartphone mounted on the windshield records the sensor data. The data used for this experiment was collected over various roads of Greenville, Spartanburg, Clemson, and Columbia area in South Carolina, USA. The data collection was performed using multiple cars. During data collection, the speed and placement of the vehicles when they pass over the road damage were also taken into consideration to produce as much diverse data as possible. The vehicles do not need to maintain constant speeds during data collection, but the final data set should contain data that covers almost the entire common speed range, such that the trained machine-learning models are free from the influence of

vehicle speed, and the severity-estimation algorithm can have enough data to calculate the necessary statistics. The different types of road damage covered during data collection were bumps, construction joints, cracks, spall, and normal road surface. The details of the collected 1378 data points can be seen in Table 1.

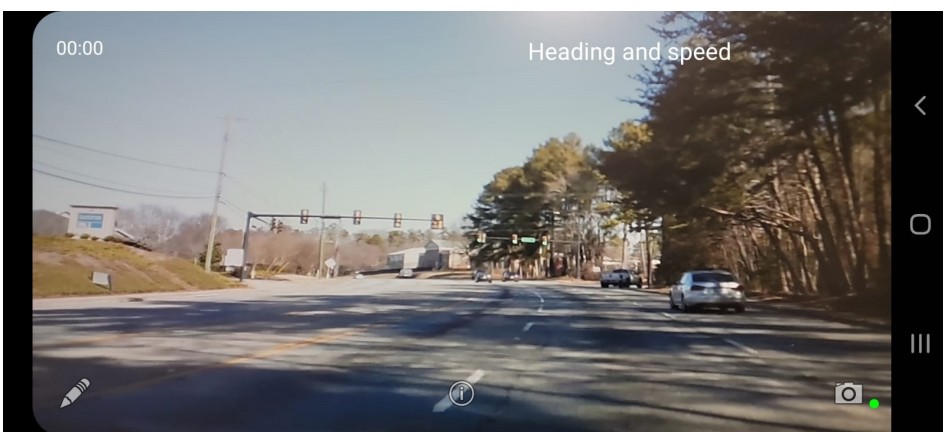

**Figure 9.** Screen view of the data-collection app.

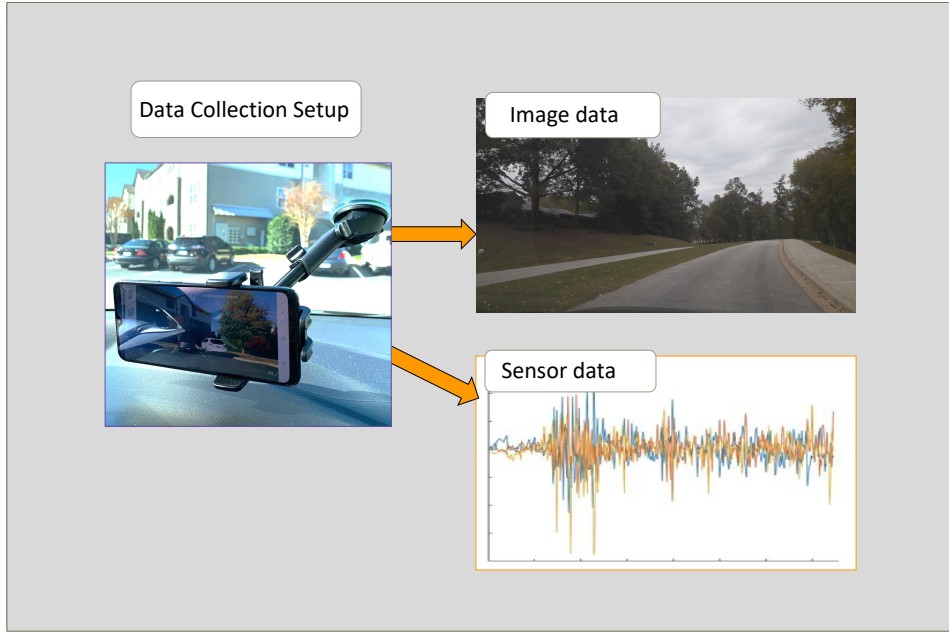

**Figure 10.** Data-collection example.

**Table 1.** Count of data across road-damage types.

| Class | Count | Class | Count |
|-------|-------|-------|-------|
| Bump | 39 | Construction Joint | 156 |
| Crack | 721 | Undamaged | 310 |
| Spall | 152 | | |

The smartphone optimizes the use of sensors, changing the frequency of data recording; hence, not all the collected data are usable. The usable data points are the ones with a sampling frequency of 60 Hz or higher. The details of the usable motion data for detection are presented in Table 2.

**Table 2.** Count of acceleration data across classes.

| Class | Count |
| --- | --- |
| Damaged | 378 |
| Undamaged | 310 |

The images from [34], one of the largest road-damage data sets, pothole data set [35], and images extracted off the video frames from the data recording were used for the vision-based YOLOv5 model as well. After careful filtering and processing, 1000 images were retained. Data-augmentation techniques, such as flip in all four directions, rotate by 90° left and right, and 20% crop, were implemented on the original images. The images were increased to 3000 after application of the augmentation techniques. This process introduced variations and increases the size of the data set. Table 3 shows the breakup of images into classes—potholes and cracks—after the augmentation techniques.

**Table 3.** Count of images across classes after augmentation.

| Class | Count |
| --- | --- |
| Potholes | 893 |
| Cracks | 767 |
| Total Annotations | 1660 |
| Total Images after Augmentation | 3000 |

*4.2. Experimental Result and Analysis*

4.2.1. Results of Vision-Based Road-Surface Detection

YOLOv5 was trained extensively for the road-damage types, including potholes and cracks, for 200 epochs. The model performed well on the validation data set with an accuracy of 87.5% mean average precision (MAP). The MAP improvement over the epochs is shown in Figure 11. The MAP gradually improved as the training progressed initially, and eventually almost stopped changing, which indicates that the model was trained successfully. Figure 12 shows the training and validation loss over the epochs. Both the training loss and the validation loss dropped to close to zero at the end of the training, meaning that the model was not encountering the overfitting issue.

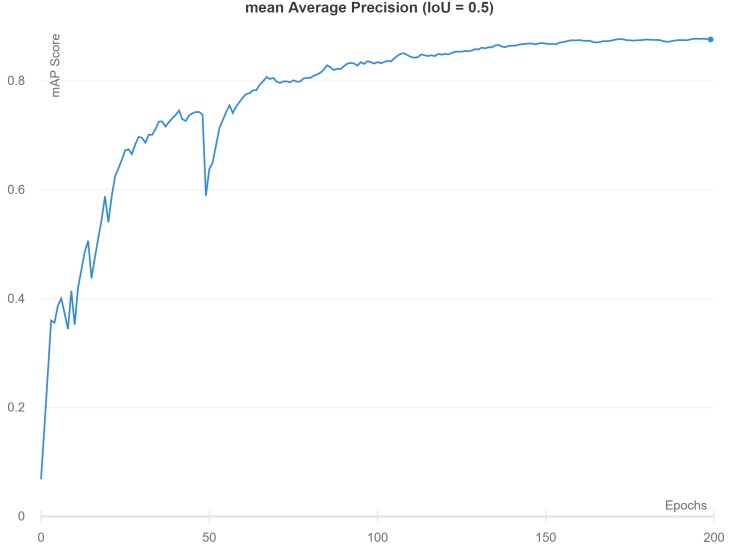

**Figure 11.** YOLOv5 mean average precision over epochs.

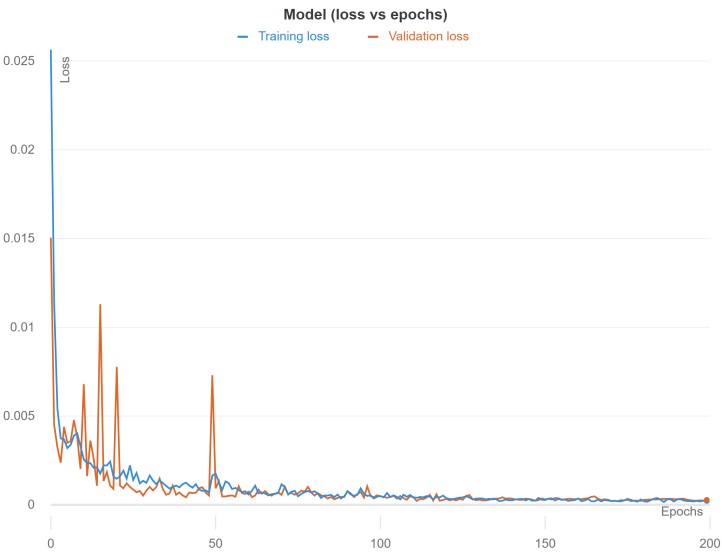

**Figure 12.** YOLOv5 loss over epochs on the training and validation data set.

Figures 13 and 14 show the confusion matrix of the inferences on the training and testing data set, respectively. This matrix compares the actual target values with those predicted by our YOLOv5 model. The rows represent the predicted values of the target variable. The column represents the ground truth of the target variable. Tables 4 and 5 provide the summary of the trained model accuracy on the training and testing data set, respectively. The performance numbers were obtained when the threshold of intersection over union (IOU) was set to 0.5 for successful detection. The results show that the trained model never confuses cracks and potholes. On the training dataset, the model did not miss hardly any detection. On the testing dataset, the model did not miss hardly any potholes, but failed to detect 22% of the cracks. On both the training and the testing datasets, the model generated some false positives when there was no road damage. The calculated F1 score is 0.853, which is better than the performance of the networks for the same purpose in [19–24]. Overall, the results show that the trained YOLO model is especially good at telling cracks from potholes, but can sometimes make mistakes in distinguishing undamaged roads from damage.

**Table 4.** Accuracy results of training data set.

| Class | Annotations | Precision | Recall | mAP@0.5 IOU |
|---|---|---|---|---|
| Potholes | 1826 | 0.898 | 0.999 | 0.997 |
| Cracks | 1621 | 0.999 | 0.984 | 0.996 |
| Total | 3447 | 0.994 | 0.991 | 0.997 |

**Table 5.** Accuracy results of testing data set.

| Class | Annotations | Precision | Recall | mAP@0.5 IOU |
|---|---|---|---|---|
| Potholes | 779 | 0.932 | 0.893 | 0.937 |
| Cracks | 660 | 0.888 | 0.730 | 0.813 |
| Total | 1439 | 0.910 | 0.812 | 0.875 |

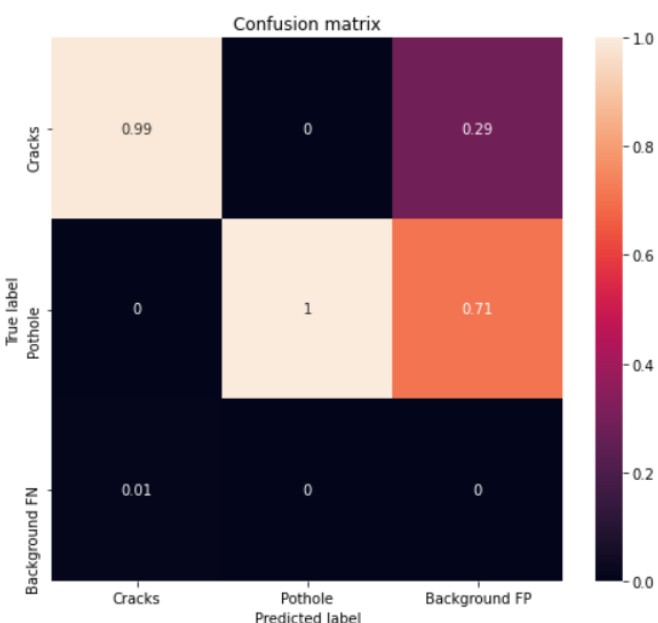

**Figure 13.** Confusion matrix—YOLOv5 model on the training data set.

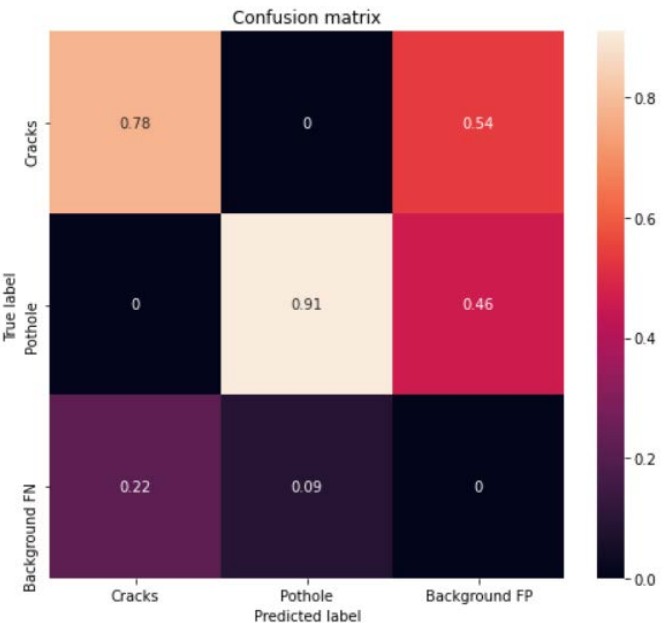

**Figure 14.** Confusion matrix—YOLOv5 model on the testing data set.

The precision vs. recall curve for the two classes, potholes and cracks, and the overall detection is as shown in Figure 15. The model was able to give 81% MAP on cracks, and 91% MAP on potholes, after training for 200 epochs. It can be seen that all three curves are very close to the top right corner, which indicates that the trained model can be an effective predictor. Figure 16 shows the model inference on some of the images from the validation data set. The two classes potholes and cracks were inferred well by our model.

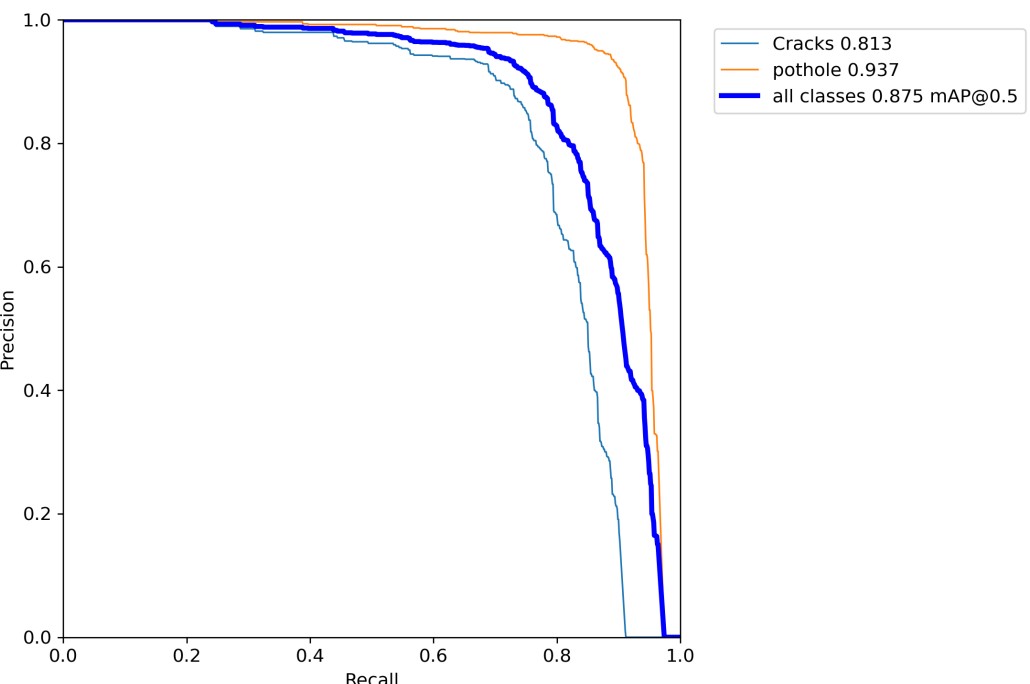

**Figure 15.** YOLOv5 precision-recall curve.

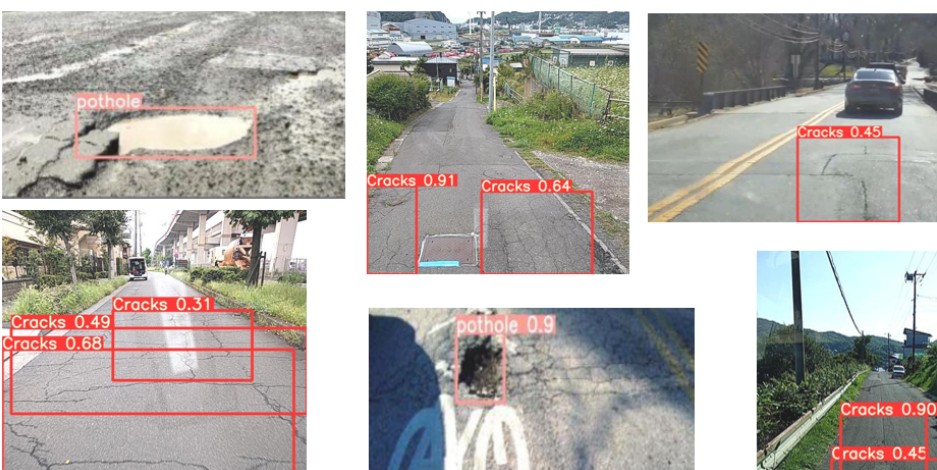

**Figure 16.** YOLOv5 inference on validation images.

### 4.2.2. Results of Motion-Based Road-Damage Detection

The LSTM model was trained on a Palmetto cluster on GPU nodes. The LSTM model was tuned well to obtain the necessary results. Table 6 gives the details on the tunable parameters used for the training. Figure 17 shows the prediction accuracy over epochs during training and Figure 18 shows how the loss reduced over epochs. It can be seen that the training was stopped when the model's performance almost stopped increasing on both the testing dataset and validation dataset and started to show differences between them. This indicates that the model was trained well before an overfitting issue started to appear. The trained LSTM model gives an accuracy of 94% in classifying the damage and normal road on the testing data set, performing better than models for similar purposes in [7–12].

**Table 6.** Tuning parameters of LSTM.

| Parameter Name | Parameter Used |
|---|---|
| Optimizer | Adam |
| Loss regularization | L2 (L2 loss used is 0.015) |
| Learning rate | 0.005 |
| Batch size | 64 |

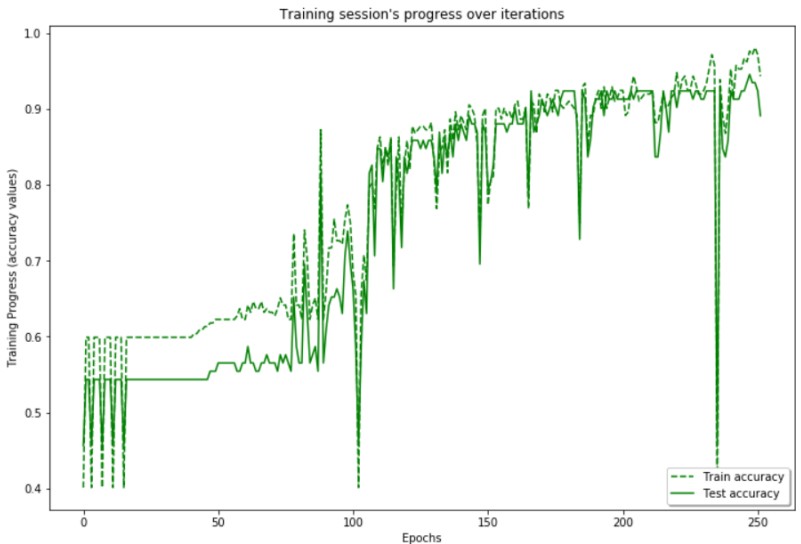

**Figure 17.** LSTM classification accuracy over epochs.

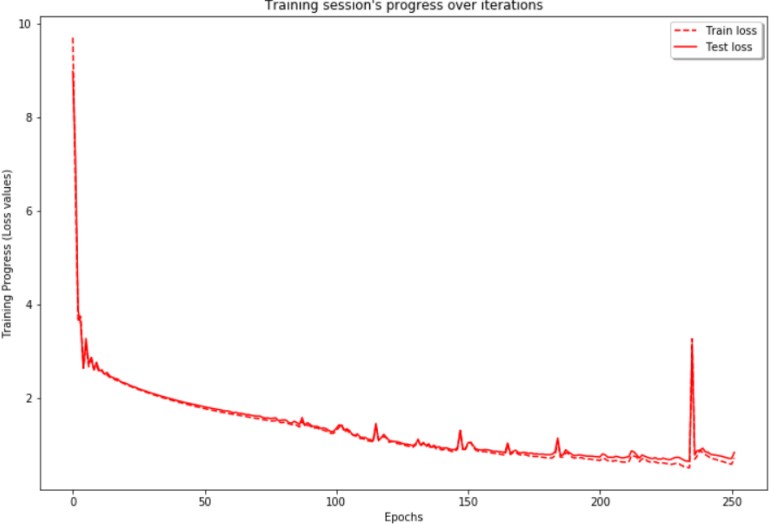

**Figure 18.** LSTM classification loss over epochs.

Figures 19 and 20 show the confusion matrix of the model on training and testing data sets, respectively. The trained LSTM model is able to achieve a 99% precision for detecting undamaged roads and 96% precision for detecting damaged roads on the training dataset, and 96% precision for detecting undamaged roads and 90% precision for detecting damaged roads on the testing dataset. The model generated very few false positives on both datasets. Overall, the LSTM model performs very well in telling damaged road surfaces from undamaged ones and can be a very good compensation for the trained YOLO model.

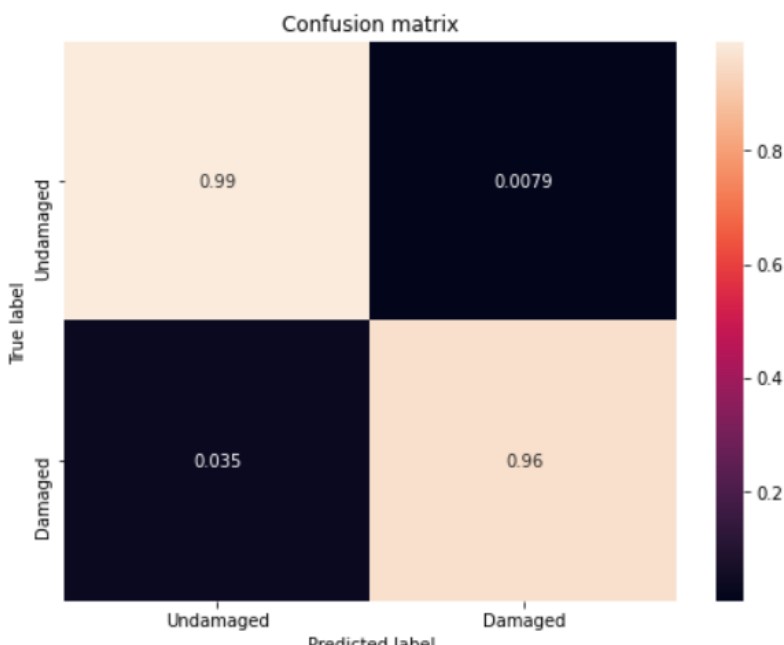

**Figure 19.** Confusion matrix—LSTM model on the training data set.

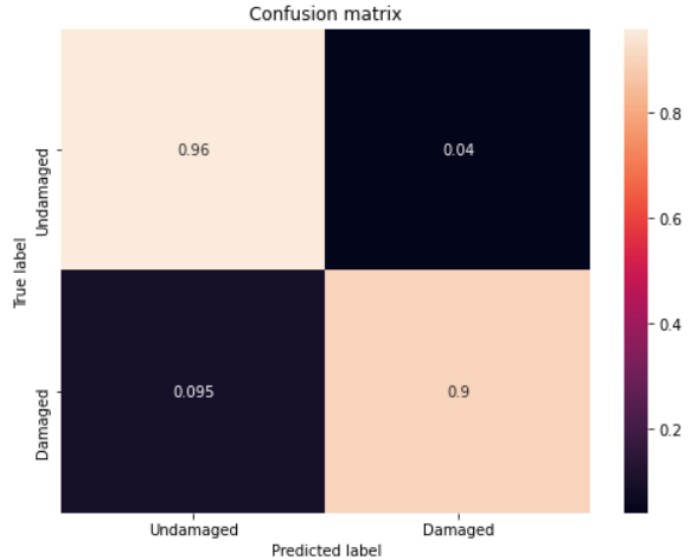

**Figure 20.** Confusion matrix—LSTM model on the testing data set.

### 4.2.3. Results of Cloud-Based Collaborative Fusion

Figure 21 shows a screenshot of the web page displaying the results of the clustering with the top three damage types and their corresponding confidence with the address of the damage. The web page enables concerned authorities to view the road damage reported by the users with the help of the mobile application. The web page comprises a map on which the locations with road damage are marked with different types of markers. The markers are of different shapes to distinguish between the highest reported type of damage in that location. There are a total of five different shapes for markers for the current two types of damage and three other types of damage to be included in the future. The marker shape and related damage type are as follows:

- Star: pothole;
- Square: cracks;
- Others: reserved;

Further, there is the ability to display the severity of the damage at a particular location, and the markers have five different colors. The five colors range from a dark green shade displaying less severe damage to a dark red color displaying severe damage. The web-page viewer can click on the markers to know more details about the damage, such as type, confidence level, and location details. For example, the location shown in the figure has 33 different damage reports with high confidence after clustering, and the most frequent damage type is potholes, which constitutes 24 out of the 33 reports, and, thus, the location most likely has potholes on the road surface. When comparing our cloud-based fusion solution with other methods, our method has the advantages of tolerating false predictions and low requirements for single-vehicle data.

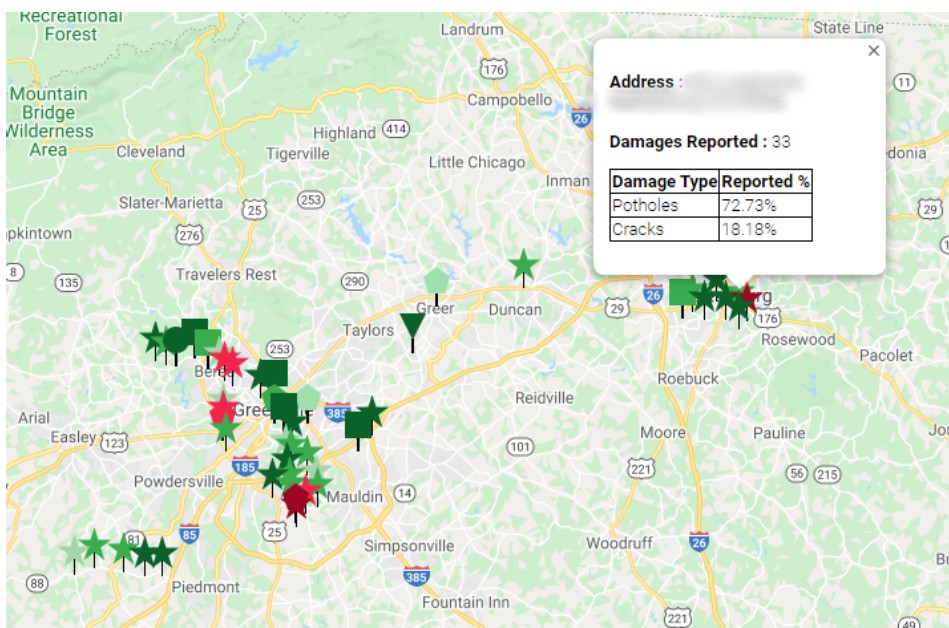

**Figure 21.** User interface web page.

### 4.2.4. Results of Cloud-Based Road-Damage Severity Estimation

In this section, we use data collected from 19 different damage locations to perform line fitting to build the mapping between vehicle vertical acceleration and road-damage depth. The data points used for line fitting are marked as blue points in Figure 22. Data from the other six damaged locations are used to test the accuracy of the road-damage estimation, which is marked as orange points in Figure 22. It can be seen that the testing data also match with the fitted line nicely.

To better demonstrate the accuracy of the road-damage severity estimation, an error analysis was also conducted. A detailed comparison between measured damage depth and estimated damage depth for the six testing locations can be found in Table 7. The average error is 1.79 cm based on the testing data, with a minimum error of 0.82 cm and a maximum error of 2.92 cm. Overall, the road-damage severity estimation method using the worst-case vertical acceleration and line-fitting technique can work effectively.

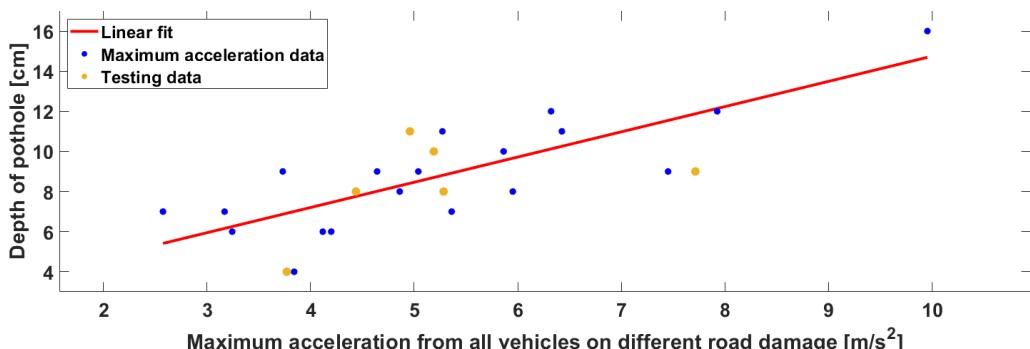

**Figure 22.** Damage-severity estimation validation.

**Table 7.** Road-damage-severity estimation error.

| Estimated data [cm] | 8.82 | 11.88 | 8.7 | 8.41 | 7.76 | 6.92 |
|---|---|---|---|---|---|---|
| Measured data [cm] | 8 | 9 | 10 | 11 | 8 | 4 |
| Error [cm] | 0.82 | 2.88 | 1.3 | 2.59 | 0.24 | 2.92 |

## 5. Conclusions

This paper focuses on the problem of monitoring road-surface conditions for detecting possible road damage. A deep-learning and cloud-based collaborative method is proposed. The method utilizes both vehicle-motion data and vision data collected by cellphones to achieve reliable and accurate detection while being cost-effective. Deep-learning-based techniques including YOLOv5 and LSTM were trained for this task. The cloud fuses detecting results from various vehicles to generate a map with credible road-damage information for drivers. The performance of the proposed method was proven in realworld experiments.However, the work can still be improved in the following aspects: The current dataset we use to train the deep-learning models can be expanded to be more comprehensive. In the future, we will collect data from more weather and lighting conditions and vehicle types. The current severity-estimation method is mainly based on motion data. In our future work, we will combine the current estimation method with information from vision data to achieve the estimation of the 3-D dimensions of road damage. In addition, the current method does not output the pavement-quality index, which is a standard index for evaluating road-surface conditions. Our future work will create an algorithm to estimate this from vehicle vertical acceleration and speed. The current cloud-based fusion algorithm has its limitations as well. Firstly, it is not very time-efficient and may have some duplicated predictions in the cloud. In the future, we will investigate more efficient fusion algorithms that can better process the duplicated and redundant information from the cloud. Secondly, the current fusion algorithm directly fuses the prediction results from cellphones, which does not fully utilize the computational power of the cloud server. Our future work will offload computational tasks from cellphones and run more complex and precise deep-learning models on the cloud to further improve prediction accuracy. Lastly, the current cloud is centralized, which is not ideal for a large area, since the communication delay increases and fusion speed drops as the map grows. Our future work will improve the cloud structure to a distributed one to accommodate large areas.

**Author Contributions:** Conceptualization, G.C. and Y.J.; Data curation, D.N., V.N.B. and L.H.; Formal analysis, A.R.; Funding acquisition, G.C. and Y.J.; Methodology, A.R., D.N., L.G. and R.W.; Project administration, G.C. and Y.J.; Resources, G.C. and Y.J.; Software, A.R., D.N., V.N.B., R.W. and L.H.; Supervision, L.G., G.C. and Y.J.; Validation, R.W.; Writing—original draft, A.R.; Writing—review & editing, L.G. All authors have read and agreed to the published version of the manuscript.

**Funding:** This study is based upon work supported by the Center for Connected Multimodal Mobility ($C^2M^2$) (a US Department Transportation Tier 1 University Transportation Center) headquartered at Clemson University, Clemson, South Carolina, US. Any opinions, findings, conclusions, or recommendations expressed in this paper are those of the authors and do not necessarily reflect the views

of the Center for Connected Multimodal Mobility, and the U.S. government assumes no liability for the contents or use thereof.

**Institutional Review Board Statement:** Not applicable.

**Informed Consent Statement:** Not applicable.

**Data Availability Statement:** Not applicable.

**Conflicts of Interest:** The authors declare no conflict of interest.

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
