# Peer review of "Cloud-Based Collaborative Road-Damage Monitoring with Deep Learning and Smartphones"

_sustainability, doi:10.3390/su14148682_

Round 1

Reviewer 1 Report

L62: Need to specify what road conditions to be monitored.

L78: Why mobile phones are able to monitor traffic-related information continuously if the users do not use or carry mobile phones when travelling on the road?

L81: What are the exactly road condition the cellphones are to measure? Is it roughness? How to use those imbedded INS to determine road condition?

L218: How did you measure 3D of road damage?

When measuring road damage by vehicle vertical acceleration, where is the cell phone located? How to ensure the vehicle to run in a constant speed to get consistently collected data?

Vehicles wander on the road. How to ensure the vehicle collect the data for the entire pavement section? Vehicle may not cross pothole.

Table 1: It is unclear how each damage type is determined by images.

It appears that the authors did not have a good understanding on the difference between fatigue crack and alligator crack. Alligator crack is bottom up fatigue cracking.

Section 3.2.4: It appears that authors came up with their own severity index. The authors would need to check the roadway practice. Do they use the same metrics and indicators as what you proposed? Otherwise, it would not be practical for agencies to use those data.

The authors are encouraged to revise the method description to be more concise and clear manner.

Author Response

Dear Reviewer,

Thank you very much for your comments on our paper “Cloud-based Collaborative Road Surface Monitoring with Deep Learning and Smartphones”. The revisions have been highlighted in blue in the new manuscript. In addition, the responses to the comments are presented in the document attached.

Thank you very much for your comments again and we look forward to hearing from you soon!

Best regards,

Akshatha Ramesh, Dhananjay Nikam, Venkat Narayananm, Longxiang Guo, Rongyao Wang, Leo Hu, Gurcan Comert and Yunyi Jia

Reviewer 2 Report

·        Author proposed deep learning methods in order to overcome the poor and damaged road conditions by using smartphones data. The device solution is cost effective as well as an ad-hoc approach.

·        Paper organization section is missing in the end of introduction of section. Briefly describe the section and subsection of your whole menu script in one paragraph. Add this paragraph in the end of the introduction section.

·        There is no literature review section in this research. In order to prove that your research is beyond state-of-the-art and experiment results are more effective, you have needed to add a literature review section must. Author should present a table of comparison of previous published articles with this manuscript and show the importance of this menuscript why it should be consider for publication although there are already many good soultions have been developed by researchers.

·        What are the challenges and gaps of research? Author should explain challenges and gaps in this area of research in a separate section.

·        What are the limitations of the research? Have you compared your results with already published articles? If yes! Then present it in tabular form.

·        Try to avoid heading after heading such as:

·        3. Experiment Results 247

·        3.1. Experimental setup and data collection 248

·        3.1.1. Setup and application development

·        Add some text between all these headings and subheadings. Minor spell checkers and grammatical errors are required to check.

Author Response

Dear Reviewer,

Thank you very much for your comments on our paper “Cloud-based Collaborative Road Surface Monitoring with Deep Learning and Smartphones”. The revisions have been highlighted in blue in the new manuscript. In addition, the responses to the comments are presented in the attached document.

Thank you very much for your comments again and we look forward to hearing from you soon!

Best regards,

Akshatha Ramesh, Dhananjay Nikam, Venkat Narayananm, Longxiang Guo, Rongyao Wang, Leo Hu, Gurcan Comert and Yunyi Jia

Round 2

Reviewer 1 Report

Comments are well addressed. No more comments.